# Surgery-Related Muscle Loss after Pancreatic Resection and Its Association with Postoperative Nutritional Intake

**DOI:** 10.3390/cancers15030969

**Published:** 2023-02-03

**Authors:** Rianne N. M. Hogenbirk, Judith E. K. R. Hentzen, Willemijn Y. van der Plas, Marjo J. E. Campmans-Kuijpers, Schelto Kruijff, Joost M. Klaase

**Affiliations:** 1Department of Surgery, University Medical Center Groningen, University of Groningen, 9713 GZ Groningen, The Netherlands; 2Department of Surgery, Amsterdam University Medical Center, 1105 AZ Amsterdam, The Netherlands; 3Department of Gastroenterology and Hepatology, University Medical Center Groningen, University of Groningen, 9713 GZ Groningen, The Netherlands

**Keywords:** surgery-related muscle loss, muscle wasting, ultrasound, POCUS, pancreatic surgery, pancreatic cancer, protein intake, nutrition

## Abstract

**Simple Summary:**

Acute muscle loss is a well-known phenomenon in critically ill patients; however, little is known about clinically relevant surgery-related muscle loss (SRML) after major surgical procedures and its association with in-hospital nutritional intake. The aim of this prospective study was to measure the occurrence of postoperative surgery-related muscle loss and to study its association with postoperative nutritional intake in patients who underwent pancreatic surgery for malignant diseases. We found that SRML occurred in 38% of the patients and was associated with low nutritional intake. This study emphasizes the urgency of more intensive dietary support in the early postoperative phase after pancreatic surgery.

**Abstract:**

To study the occurrence of surgery-related muscle loss (SRML) and its association with in-hospital nutritional intake, we conducted a prospective observational cohort study including patients who underwent pancreatic surgery because of (suspected) malignant diseases. Muscle diameter was measured by using bedside ultrasound 1 day prior to surgery and 7 days postoperatively. Clinically relevant SRML was defined as ≥10% muscle diameter loss in minimally one arm and leg muscle within 1 week after surgery. Protein and caloric intake was measured by nutritional diaries. The primary endpoint included the number of patients with SRML. Secondary endpoints included the association between SRML and postoperative nutritional intake. Of the 63 included patients (60.3% men; age 67.1 ± 10.2 years), a total of 24 patients (38.1%) showed SRML. No differences were observed in severe complication rate or length of hospital stay between patients with and without SRML. During the first postoperative week, patients with clinically relevant SRML experienced more days without any nutritional intake compared with the non-SRML group (1 [0–4] versus 0 [0–1] days, *p* = 0.007). Significantly lower nutritional intake was found in the SRML group at postoperative days 2, 3 and 5 (*p* < 0.05). Since this study shows that SRML occurred in 38.1% of the patients and most of the patients failed to reach internationally set nutritional goals, it is suggested that more awareness concerning direct postoperative nutritional intake is needed in our surgical community.

## 1. Introduction

Acute muscle loss is a well-described phenomenon in critically ill intensive care unit (ICU) patients [1,2,3,4]. During the past decade, the literature analyzing the role of immobilization and nutritional demands in relation to muscle wasting of these ICU patients has considerably expanded [5,6,7]. However, little is known about muscle loss and the role of nutrition in the non-critically ill postoperative surgical patient, and since previous reports have shown an association between acute muscle loss with quality of life and survival in postoperative patients, identifying the extent of postoperative muscle loss and potentially the prevention of postoperative muscle loss is of high importance [8,9,10].

Existing research recognizes the metabolic changes induced by surgical trauma [11,12]. Especially after major surgery, the surgical stress response—which is characterized by neurohormonal, systemic and immunologic pathways—endeavors to increase plasma and heart-minute volume, to improve oxygen uptake and to mobilize energy reserves to accelerate postoperative recovery [12,13,14]. Additionally, for the production of acute phase enzymes, an increased excretion of cortisol, growth hormones and catecholamines leads to a hypermetabolic state characterized by hyperglycemia and protein catabolism, which results in muscle wasting [12,13,14,15,16]. In order to maintain muscle mass and prevent protein catabolism in this direct postoperative phase, the European Society for Clinical Nutrition and Metabolism (ESPEN) stated that adequate nutritional intake of 1.5 g protein per kilogram (kg) bodyweight and 15–20 kilocalories (kcal) per kg bodyweight is of high importance [7,17,18].

We hypothesize that patients with low energy and protein reserves, such as most patients with pancreatic malignancies [19], have a diminished physiological reserve capacity to satisfy the increased physical demands of the surgical stress response, which results in increased muscle wasting when postoperative intake is inadequate. Therefore, the aim of this study is to gain insight in the amount of clinically relevant surgically related muscle loss (SRML) after pancreatic surgery and to investigate its association with postoperative protein and energy intake.

## 2. Materials and Methods

### 2.1. Study Design

This study is part of the MUSCLE POWER study, an observational prospective cohort study aiming to assess risk factors and clinical impact of clinically relevant SRML in patients undergoing major open abdominal surgery at the University Medical Center Groningen (UMCG). Patients scheduled for major open pancreatic surgery based on an underlying suspected (pre)malignant disease were included in the period of May 2019 to June 2021. Because of hospital policy limiting access of researchers due to peak-incidence of COVID-19, inclusion of patients was temporarily stopped between February 2020 and September 2020.

The study protocol was approved by UMCG’s medical ethics committee (METc2018/361, version 3.0, 21 January 2019), registered within the International Clinical Trials Registry Platform (201800445, NL7505, version 1.0, 7 February 2019) and was published previously [20]. The study was performed in accordance with the ethical standards laid down in the 1964 Declaration of Helsinki and its later amendments [21].

### 2.2. Data Collection

#### 2.2.1. Baseline and Perioperative Characteristics

For each patient included in the study, demographic data including age, sex, weight (kg), height (cm), smoking status, comorbidities and the American Society of Anesthesiologists score [22] were prospectively collected from electronic patient charts and supplemented during patient interview. Data on preoperative nutritional status and weight loss was collected by using the Patient-Generated Subjective Global Assessment Short Form (PG-SGA SF) [23]. This validated questionnaire for nutritional assessment focuses on four elements: dietary intake change, gastrointestinal symptoms, short-term weight loss and changes in functional capacity. Patients with a PG-SGA score of 4 or higher were considered as malnourished [23]. Surgical parameters such as type of operation (i.e., pylorus preserving pancreatoduodenectomy [PPPD], Whipple’s procedure, distal pancreatectomy), operation time and intra-operative blood loss were collected from surgical reports. Collected postoperative characteristics included all complications clustered according to Clavien–Dindo scores (with severe complications defined as grade ≥ 3) [24,25], specific complications related to pancreatic surgery clustered conform the International Study Group of Pancreatic Surgery (ISGPS) definition (i.e., delayed gastric emptying grade B or C (>8–14 days of nasogastric tube [NGT] required or reinsertion NGT after 7 days), postoperative pancreatic or biliary fistula with therapeutic consequences (grade B or C), post-pancreatectomy hemorrhage with therapeutic consequences (grade B or C), chylous leakage requiring artificial nutrition or invasive therapy (grade B or C) [26,27,28,29], length of hospital stay, length of ICU-stay, mortality and readmission rate within 90 days after discharge.

#### 2.2.2. Muscle Mass

Muscle mass was assessed by bedside Point-of-Care Ultrasound (POCUS) (Philips, FUS6882 Lumify L12-4) by measurement of the following three muscles bilaterally: m. biceps brachii, m. rectus femoris and m. vastus intermedius. Patients were positioned in supine position on the bed with arm and leg muscles relaxed. Muscle measurements were performed by three researchers (JH, WP, RH) who were trained by a musculoskeletal radiologist (AV) according to a predefined standardized protocol [20]. For each muscle anterior/posterior diameter was measured three times, after which the mean of the three anterior/posterior diameter measurements was used in analyses. The location of measurements of the m. biceps brachii were performed at two-thirds the length between the acromion and elbow fold with the elbow passively extended. Measurements of the m. rectus femoris and m. vastus intermedius were performed halfway between the spina iliaca anterior superior and proximal border of the patella. The measure points were marked with a waterproof marker to ensure fixed points of measurements. Measurements were conducted 1 day prior to surgery (i.e., baseline measurements) and repeated 3, 7, and 10 days after surgery (Figure 1). A decline of ≥10% diameter of at least one arm muscle and one leg muscle at the seventh postoperative day was considered as clinically relevant SRML in this study.

#### 2.2.3. Nutritional Intake

All postoperative nutritional intake through oral, enteral and parenteral nutrition was recorded by the patients, nurses or responsible researchers in a nutritional diary up to the seventh day after surgery. Thereafter, consumed protein and energy intake was calculated by use of a nutrition calculator application (Mijn Eetmeter, Stichting Voedingscentrum Nederland, Rotterdam, The Netherlands). Protein intake was assessed as grams (g) protein per kilogram (kg) bodyweight per day. Energy intake was assessed as kilocalories (kcal) per kg bodyweight per day.

### 2.3. Study Endpoints

Primary endpoint of this study was the number of patients presenting with clinically relevant SRML after 7 postoperative days. Secondary endpoints included the association between clinically relevant SRML and postoperative nutritional protein and energy intake, complication rate, length of hospital stay and readmission rate.

### 2.4. Statistical Analysis

Nutritional intake was calculated per postoperative day separately. When patients were discharged within 1 week after surgery, measurements of nutritional intake were assessed upon the day prior to discharge, and muscle diameter was measured at the day of discharge. When the last muscle measurements were performed before the sixth day postoperatively, patients were excluded from analysis of clinically relevant SRML.

Continuous variables are presented as mean with standard deviation (SD) or as median with interquartile range (IQR) based on distribution. Categorical variables are described as count (n) and percentage (%). Independent samples Student’s *t* tests, Mann–Whitney U tests, Pearson’s χ^2^ test and Fisher’s exact test were used to compare patients with and without clinically relevant SRML.

Pearson’s product moment correlation coefficient was used for testing of correlations between patients’ baseline muscle diameter measurements and relative muscle loss after 1 week postoperatively. *p*-values below 0.05 were considered statistically significant. All statistical analyses were performed using the Statistical Package for the Social Sciences (SPSS 23.0, SPSS, Chicago, IL, USA).

## 3. Results

### 3.1. Patients

During the study periods between May 2019 to March 2020 and September 2020 to June 2021, a total of 86 patients scheduled for open abdominal pancreatic surgery were assessed for eligibility of inclusion. Of these patients, a total of 15 patients declined participation in the study prior to surgery, 7 patients were excluded after inclusion because no major open abdominal surgical resection was performed due to progressive disease and 1 patient passed away prior to the seventh postoperative day. The remaining 63 patients who underwent pancreatic surgery (mean age 67 ± 10.1; 38 [60.3%] male) were included in the analyses. Baseline characteristics of patients with and without the presence of clinically relevant SRML are presented in Table 1.

### 3.2. Presence of Clinically Relevant SRML

A total of 24 patients (38.1%) showed clinically relevant SRML at the seventh postoperative day. Within the entire cohort, preoperative weight loss of more than 5% in the prior 6 months was observed in 28 patients (44.4%). Patients without clinically relevant SRML presented significantly more often with preoperative malnutrition (PG-SGA ≥ 4) compared with patients with clinically relevant SRML (12 [30.8%] versus 2 [8.3%], *p* = 0.038). No other statistically significant differences were found between patient characteristics in both groups (Table 1). A graphical representation of the dynamic changes of postoperative muscle thickness per measured muscle in both arms and legs compared between patients with and without the presence of clinically relevant SRML is presented in Figure 2.

### 3.3. Postoperative Course

Table 2 shows an overview of all intraoperative and postoperative characteristics of the entire cohort. Indication for surgery in the majority of the patients was due to a histological diagnosis of pancreatic adenocarcinoma (34 patients [54.0%]). The majority of patients underwent a PPPD procedure (43 patients [68.3%]). Patients with clinically relevant SRML tended to have more often an intraductal papillary mucinous neoplasm (IPMN) at final pathology examination compared with the no-SRML group (5 [20.8%] versus 2 [5.1%], *p* = 0.054).

Median length of ICU stay and total postoperative hospital stay was 1 (0–1) day and 12 (10–17) days, respectively. A total of 14 patients (22.2%) suffered from severe complications (i.e., Clavien–Dindo grade three or higher). No significant differences in severe complication rate were found between patients with and without clinically relevant SRML (6 [25.0%] versus 8 [20.5%], *p* = 0.677). Neither were significant differences regarding length of hospital stay, specific pancreatic-surgery-related complications and readmission rate observed between the groups with and without clinically relevant SRML.

### 3.4. Nutritional INTAKE

Increase in median protein and energy intake per kg bodyweight during the first postoperative week was observed in both, clinically relevant SRML and non-SRML, groups. However, during the first postoperative week, patients with clinically relevant SRML experienced significant more days without any nutritional intake compared with the non-SRML group (median 1 [0–4] days versus 0 [0–1] days, *p* = 0.007). On the first, second and third postoperative day, a significant higher number of patients with clinically relevant SRML did not receive any oral, enteral or parenteral protein or energy intake compared with patients without clinically relevant SRML (first day: 13 [54.2%] versus 10 [25.6%], *p* = 0.022; second day: 10 [41.6%] versus 3 [7.7%], *p* = 0.001; third day: 7 [29.2%] versus 2 [5.1%], *p* = 0.008, respectively).

Patients with clinically relevant SRML had significantly lower median protein intake compared to the non-SRML-group at postoperative day 2 (0.06 [0–0.71] versus 0.45 [0.10–0.81] g/kg, *p* = 0.029), day 3 (0.25 [0–1.04] versus 0.96 [0.28–1.20] g/kg, *p* = 0.029) and day 5 (0.32 [0.02–1.54] versus 0.99 [0.41–1.55] g/kg, *p* = 0.019). Regarding energy intake, a significantly lower intake was observed in the clinically relevant SRML group at the second postoperative day (2 [0–13.58] versus 10.91 [2.50–14.37] kcal/kg, *p* = 0.042) and at the fifth postoperative day (6.52 [1.50–21.29] versus 18.47 [5.78–28.25] kcal/kg, *p* = 0.030) (Table 3).

After 5 days, a significantly lower cumulative protein intake was observed in the SRML group compared to the no-SRML group (0.92 g/kg (0.20–4.40) protein vs. 3.32 g/kg (1.03–5.70), *p* = 0.038). A trend toward a lower cumulative energy intake was observed in the SRML group after 5 cumulative days (21.09 kcal/kg (3.65–74.26) vs. 60.31 kcal/kg [18.90–108.17], *p* = 0.053)

A total of 32 patients (50.8%) received an enteral nutritional feeding tube directly after surgery. Although not statistically significant, a trend was observed in which patients with clinically relevant SRML less often had received a direct intraoperative placed nasojejunal feeding tube compared with the group without clinically relevant SRML (9 [37.5%] versus 23 [59.0%], *p* = 0.098).

When assessing enteral (i.e., nasojejunal) nutrition solely, on the fifth postoperative day, per kg bodyweight, a significantly lower median protein and energy enteral nutrition intake by nasojejunal feeding tube was observed in patients with SRML compared with the patients without SRML (protein intake: 0 [1–0.94] versus 0.45 [0–1.25] g/kg, *p* = 0.045; energy intake: 0 [0–15.84] versus 8.9 [0–21.54] kcal/kg, *p* = 0.049).

None of the included patients received parenteral nutrition during one of the first 5 postoperative days. At the seventh postoperative day, a total of five patients (7.9%) received parenteral nutrition (3 [12.5%] with clinically relevant SRML versus 2 [5.1%] without clinically relevant SRML, *p* = 0.285).

## 4. Discussion

Our prospective study, including 63 patients who underwent open pancreatic surgery for (pre)malignancies, shows that clinically relevant SRML occurs within the first 7 postoperative days in more than one-third of the patients, and, more importantly, is associated with a reduced postoperative nutritional intake. An association between muscle diameter loss and postoperative complications could not be found. The association between nutritional intake and postoperative muscle loss could be a potential lead to reduce postoperative muscle wasting.

Assessment of skeletal muscle status is often performed by use of dual X-ray absorptiometry (DXA), bioelectrical impedance analysis (BIA) or by use of segmentation of total abdominal skeletal muscle area on the third lumbar level on abdominal computed tomography (CT) scans [30,31,32,33]. In this study, POCUS was used to perform repeated measurements of skeletal muscle thickness. Previous studies have proven that measuring muscle diameter by POCUS is feasible and reliable in the supine positioned patient [34,35,36]. Additionally, a moderate to strong correlation between POCUS muscle measurements of the extremities and CT-derived Skeletal Muscle Index at the third lumbar level was previously described [37]. When comparing our results of muscle diameter measured by POCUS with the existing literature, a comparable decrease of mean muscle diameter of the m. biceps brachii of circa 3.1 cm to 2.8 cm was found in the previous literature measuring muscle diameter after 7 days of ICU admission [38,39]. However, when comparing levels of muscle diameter of the m. rectus femoris and m. vastus intermedius, baseline muscle diameter in our study seemed to be lower than in other studies including patients presenting in an acute setting [40,41,42]. This is probably explained by the cachectic state of the average preoperative pancreatic cancer patient [19]. As a matter of fact, 44.4% of our included patients presented with minimally 5% weight loss in the 6 months prior to surgery. When comparing our measured baseline muscle diameter with other studies including elderly patients or patients with sarcopenia, chronic kidney disease or malignancies, we found comparable values of (preoperative) muscle diameter [43,44,45,46,47,48,49].

Although mostly described in ICU patients, the literature concerning risk factors for muscle wasting after surgery is increasing. For example, in a previous prospective study including a total of 110 patients, where muscle mass was measured by use of computed tomography (CT), a decrease of ≥10% of total abdominal muscle area was found in 32% of the patients within 1 week after gastrectomy and was significantly associated with older age and diabetes mellitus [8]. These risk factors are consistent with another study conducted by Van Wijk et al., where a mean decrease in total psoas area of 7.1% within 1 week after liver resection for colorectal liver metastases was found in 52% of the patients [10]. Although equal percentages of occurrence of clinically relevant surgery induced muscle loss, in contrast to these abovementioned studies, no significant association was found between muscle loss and age or diabetes in our study [8,10].

New in our study is the focus on the prospectively measured association between nutritional intake and muscle loss after pancreatic surgery. In the past years, developments regarding improvement of perioperative care by means of enhanced recovery after surgery (ERAS) programs have increased rapidly. The concept of ERAS is based on focus on early mobilization, early initiation of nutritional intake and reduction of drains and tubes (i.e., epidural pain management and nasogastric tubes) in the early postoperative phase [50,51,52]. The ESPEN highlights the increased postoperative nutritional demands and recommends early initiation of oral or enteral nutrition within 24 h after surgery [7,17,18]. Based on previous studies conducted in ICU setting with critically ill patients, the ESPEN estimated that the minimal postoperative protein and energy requirement to maintain postoperative muscle mass is 1.5 g protein and 25–30 kcal per kg bodyweight per day [7,17,18]. However, unlike previous studies [53,54,55], our study shows that a large proportion of patients does not reach this daily nutritional goal. As previously described by our study group, possible explanations could lie in patient-related factors such as postoperative discomfort, pain, nausea and gastroparesis; in health-care related factors such as missed meals and lack of focus on nutrition by doctors and nurses; or could be related to the guidelines of the ERAS programs [56].

Specific ERAS guidelines for perioperative care after pancreatoduodenectomy state that most patients tolerate normal oral intake soon after surgery and that enteral tube feeding does not necessarily confer benefit [52]. Despite this, our study illustrates that neither the patients with clinically relevant SRML nor the patients without clinically relevant SRML reached this nutritional goal. Thereby, 36.5% of the patients with clinically relevant SRML did not receive any oral, enteral or parenteral nutrition on the first postoperative day. The findings of this study underline that, despite an increased focus on ERAS-programs to enhance recovery in the specific pancreatic surgical patient, nutritional intake plays an even more crucial role in prevention of muscle loss. In this respect, ESPEN and ERAS guidelines do not combine with each other. Standardized protocols including direct intraoperative placement and caution of withdrawal of nasojejunal feeding tubes, combined with increased focus on the importance of oral intake of protein enriched products could be a manner to increase postoperative nutritional intake in these often already malnourished patients. Additionally, by providing more education on nutritional intake to patients and health-care workers during both the preoperative and postoperative phase, patients are better prepared (emotionally and physically) to receive nutritional interventions [57].

Regarding the importance of improving postoperative nutritional intake, parenteral feeding does not seem to be the solution since a large randomized controlled trial, conducted with critically ill patients (n = 4640), showed that early initiation of parenteral nutrition was correlated with a prolonged need for vital organ support and an increased incidence of infections [58]. The authors suggested that catabolism induced by higher protein demands is essential to clear intracellular microorganisms and macromolecular damaged cells in order to regenerate new tissue. It is suggested that early initiation of parenteral nutrition could induce suppression of this necessary autophagy and endogenous catabolism [58,59]. In our study, patients received parenteral feeding only after 7 days. For the pancreatic surgery patient, although the study was not designed to elaborate on the indication for early nasojejunal feeding, early enteral feeding by an intraoperatively placed jejunal feeding tube could possibly overcome the period of delayed gastric emptying and, thereby, limit clinically relevant SRML, as was suggested by the lower amount of patients with clinically relevant SRML in the group with an enteral feeding tube.

In the current study, it seems that the preoperative ‘well-nourished’ patient, i.e., the patient with a preoperative PG-SGA score below 4, is prone to lose relatively more muscle mass. An explanation could be found in the thought that patients who already suffered from severe weight loss, cannot lose more muscle mass. Or it could be explained by the fact that, in our clinic, patients who score a PG-SGA SF score ≥ 4 in the preoperative phase are referred to the dietician prior to surgery to receive preoperative nutritional support. These patients receive education and support about sufficient protein intake preoperatively and are possibly more aware of the importance of adequate nutritional intake postoperatively. Additionally, resection of the tumor that causes the cachexia could stop or reduce the hypermetabolism of muscle protein [60]. The results of this study, in combination with the observed significant association between nutritional intake and clinically relevant SRML, emphasize the importance of not only focusing on the already malnourished patient but also focusing on adequate nutritional support for all postoperative patients. Thereby, as shown in the baseline results of this study, 44.4% of the patients show >5% weight loss prior to surgery. These results are important to consider since this highlights the vulnerability of the group of pancreatic-surgery patients and speculates about the potential for optimalization by adequate nutritional support during the preoperative phase as essential part of a multimodal prehabilitation program. Future studies should focus on both the prevention of inadequate nutritional intake during the pre- and postoperative phase and on the possible long-term sequelae of clinically relevant SRML.

This study has certain limitations. The first limitation concerns the presence of peripheral edema caused by acute postoperative fluid-shifting induced by the surgical stress response [12,13,14]. As a result of the often bedridden postoperative patient, edema gathers between the subcutis and muscle fibers of the pelvis and the upper legs. This caused an increase in measured muscle diameter at the third postoperative day and therefore a possible underestimation of the group with clinically relevant SRML after 7 days. A second limitation involves the amount of pre-operative weight, and thus muscle, loss in the included patients. As pancreatic cancer patients often have major weight loss before surgery, one should be cautious with extrapolating results of this study to other ‘major abdominal surgery patients’ [19]. Additionally, the histological and surgical heterogeneity may have affected the results since PPPD, Whipple’s procedure, distal pancreatectomy and total pancreatectomy are different kinds of surgical procedures with different postoperative risks of complications and outcomes. Preferably, future research should focus on factors of influence in SRML in a more homogeneous group of patients with the same histological disease who underwent the same surgical procedure. However, the strengths of this study include the prospectively measured muscle loss and nutritional intake and the pure display of the association between nutritional intake and muscle loss in the pancreatic surgery patient.

## 5. Conclusions

In conclusion, SRML occurs in 38% of the patients within 1 week after pancreatic surgery. This is one of the first studies showing that postoperative nutritional intake after pancreatic surgery is insufficient in most of the patients and that lack of nutritional support is correlated to possibly preventable muscle loss. Since we found that neither the patients with clinically relevant SRML nor the patients without clinically relevant SRML reached the set nutritional goal, the findings of this study suggest that one should increase focus on direct postoperative nutritional or even enteral support for all patients. This is especially important when considering that postoperative nutrition could be a lead in prevention of clinically relevant SRML.

## Figures and Tables

**Figure 1 cancers-15-00969-f001:**
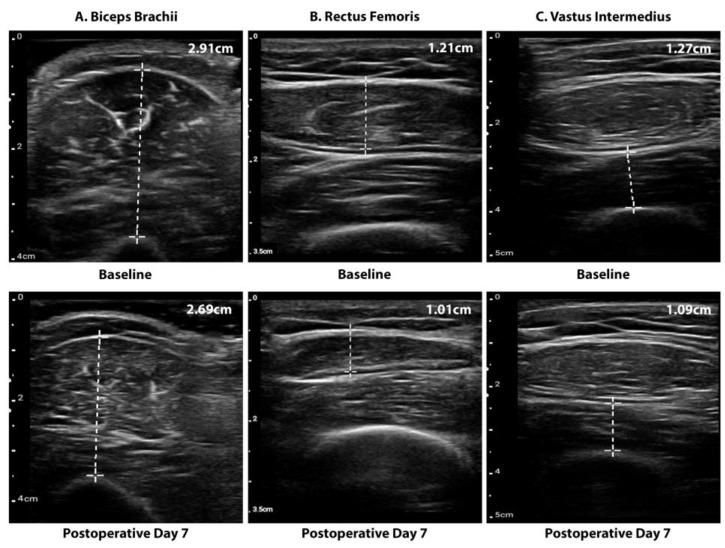
Representative images of ultrasound muscle measurements of the cross-section (anterior/posterior diameter) of (**A**) the m. bicieps brachii, (**B**) the m. rectus femoris and (**C**) the m. vastus intermedius at baseline (upper images) and after 7 days postoperatively (lower images).

**Figure 2 cancers-15-00969-f002:**
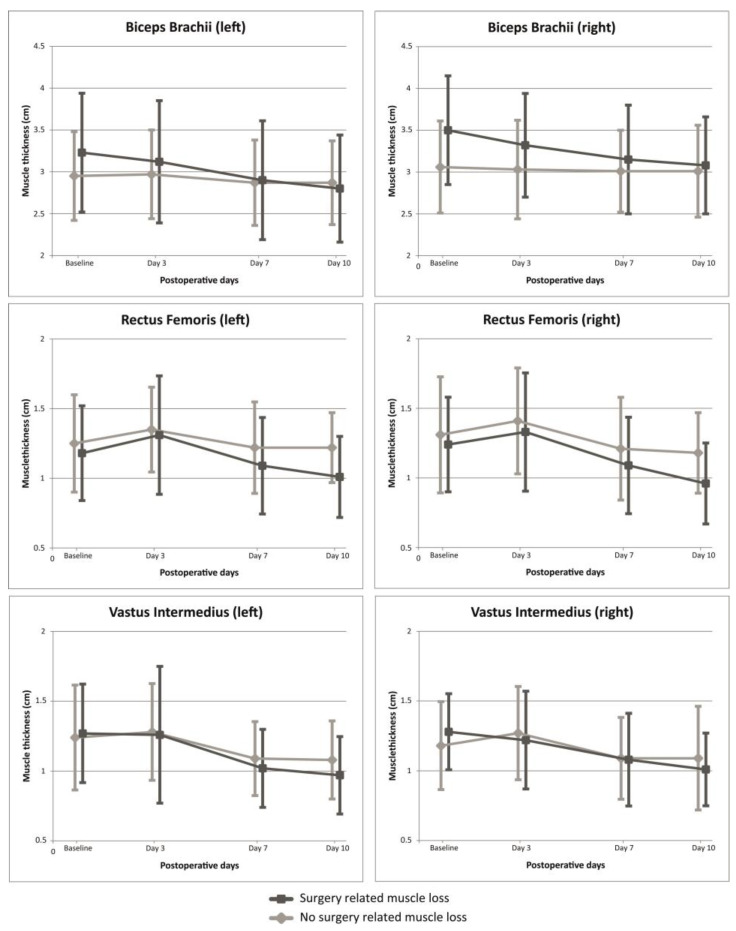
Dynamic changes of mean (SD) postoperative muscle diameter of bilateral m. biceps brachii, m. vastus intermedius and m. rectus femoris measured by ultrasound at baseline, 3, 7 and 10 days after pancreatic surgery compared between the presence of clinically relevant SRML.

**Table 1 cancers-15-00969-t001:** Patients characteristics compared between the presence of clinically relevant SRML.

Patient Characteristics	Total (n = 63)	SRML (n = 24)	No-SRML (n = 39)	*p*-Value
Sex, male	38 (60.3%)	18 (75.0%)	20 (51.3%)	0.062
Age, years	67.1 ± 10.2	68.1 ± 8.9	66.4 ± 10.9	0.536
BMI, kg/m^2^	26.3 ± 4.4	25.9 ± 4.2	26.6 ± 4.5	0.512
ASA score ≥ 3	17 (27.0%)	6 (25.0%)	11 (28.2%)	0.781
Baseline muscle cross-section, cm				
Mm. biceps brachii	3.12 ± 0.60	3.31 ± 0.67	3.01 ± 0.53	0.060
Mm. rectus femoris	1.23 ± 0.31	1.21 ± 0.34	1.28 ± 0.36	0.460
Mm. vastus intermedius	1.26 ± 0.35	1.27 ± 0.28	1.21 ± 0.33	0.431
Charlson Comorbidity Index	3 (2–4)	3 (2–4)	3 (2–4)	1
Comorbidities				
Diabetes	14 (22.2%)	4 (16.7%)	10 (25.6%)	0.405
Cardiopulmonary disease	25 (39.7%)	8 (33.3%)	17 (43.6%)	0.419
Renal comorbidity	3 (4.8%)	2 (8.3%)	1 (2.6%)	0.296
No comorbidities	22 (34.9%)	9 (37.5%)	13 (33.3%)	0.736
Smoking				
Current smoking	10 (15.9%)	4 (16.7%)	6 (15.4%)	0.892
Stopped smoking	35 (55.6%)	13 (54.2%)	22 (56.4%)	0.862
Never smoked	18 (28.6%)	7 (29.2%)	11 (28.2%)	0.935
Preoperative weight loss in 6 months				
<5% weight loss	35 (55.6%)	14 (58.3%)	21 (53.8%)	0.728
5–10% weight loss	15 (23.8%)	7 (29.2%)	8 (20.5%)	0.434
≥10% weight loss	13 (20.6%)	3 (12.5%)	10 (25.6%)	0.211
PG-SGA SF	2 (1–4)	2 (1–3.75)	3 (1–7)	0.148
PG-SGA SF ≥ 4	14 (22.2%)	2 (8.3%)	12 (30.8%)	**0.038**
Serum albumin prior to surgery	42.6 ± 3.8	42.9 ± 4.0	42.4 ±3.7	0.600
Serum albumin at discharge	32.2 ± 3.5	32.7 ± 2.8	31.9 ± 3.8	0.405

Data are presented as mean ± SD, median (IQR) or number (%). SRML = surgery-related muscle loss; BMI = body mass index; ASA = American Society of Anesthesiologists score, PG-SGA SF = Patient-Generated Subjective Global Assessment Short Form.

**Table 2 cancers-15-00969-t002:** Surgical details compared between the presence of clinically relevant SRML.

Surgical Details	Total Cohort (n = 63)	SRML (n = 24)	No-SRML (n = 39)	*p*-Value
Histological Diagnosis				
Adenocarcinoma pancreas	34 (54.0%)	11 (45.8%)	23 (59.0%)	0.310
Adenocarcinoma bile ducts	15 (23.8%)	7 (29.2%)	8 (20.5%)	0.434
IPMN	7 (11.1%)	5 (20.8%)	2 (5.1%)	0.054
Neuroendocrine tumor	3 (4.8%)	1 (4.2%)	2 (5.1%)	0.862
Other malignancy	1 (1.6%)	0 (0%)	1 (2.6%)	0.429
No malignancy	3 (4.8%)	0 (0%)	3 (7.7%)	0.164
Neoadjuvant chemotherapy	8 (12.7%)	2 (8.3%)	6 (15.4%)	0.414
Adjuvant chemotherapy	29 (46%)	10 (41.7%)	19 (48.7%)	0.586
Surgical procedure				
PPPD	43 (68.3%)	15 (62.5%)	28 (71.8%)	0.441
Whipple	10 (15.9%)	4 (16.7%)	6 (15.4%)	0.892
Total pancreatectomy	2 (3.2%)	1 (4.2%)	1 (2.6%)	0.725
Distal pancreatectomy	8 (12.7%)	4 (16.7%)	4 (10.3%)	0.458
Duration of surgery, min	469 ± 118	492 ± 122	454 ± 115	0.218
Intraoperative blood loss, mL	500 (300–750)	500 (312–819)	400 (300–750)	0.509
Surgical outcome				
Length of hospital stay, days	12 (10–17)	14 (10–19)	12 (9–16)	0.356
Length of ICU stay, days	1 (0–1)	1 (1–2)	1 (0–1)	0.140
Complications				
Comprehensive Complication Index	21 (9–30)	21 (9–33)	21 (9–30)	0.674
Clavien–Dindo ≥ 3	14 (22.2%)	6 (25.0%)	8 (20.5%)	0.677
ISGPS definition ≥ grade B or C				
Delayed gastric emptying	11 (17.5%)	5 (20.9%)	6 (15.4%)	0.527
Pancreatic fistula	14 (22.2%)	5 (20.9%)	9 (23.1%)	0.903
Hemorrhage	5 (7.9%)	2 (8.3%)	3 (7.7%)	0.889
Chylous leakage	10 (15.9%)	4 (16.7%)	6 (15.4%)	0.836
Biliary fistula	1 (1.6%)	0 (0.0%)	1 (2.6%)	0.439
Surgical re-intervention	5 (7.9%)	3 (12.5%)	2 (5.1%)	0.927
Radiological re-intervention	9 (14.3%)	4 (16.7%)	5 (12.8%)	0.672
Readmission within 30 days	9 (14.3%)	5 (20.8%)	4 (10.3%)	0.244
Readmission within 90 days	15 (23.8%)	5 (21.7%)	10 (25.6%)	0.729
Cholangitis	4 (6.3%)	2 (8.3%)	2 (5.1%)	0.257
Failure to thrive	4 (6.3%)	2 (8.3%)	2 (5.1%)	0.257
Other	7 (11.1%)	1 (4.2%)	6 (15.4%)	0.169
Mortality within 90 days	1 (1.6%)	1 (4.2%)	0	0.199

Data are presented as mean ± SD, median (IQR) or number (%). SRML = surgery-related muscle loss; IPMN = intraductal papillary mucinous neoplasm; PPPD = pylorus preserving pancreatoduodenectomy; ICU = intensive care unit; CCI = Comprehensive Complication Index; ISGPS = International Study Group of Pancreatic Surgery.

**Table 3 cancers-15-00969-t003:** Daily postoperative protein and energy intake per kilogram bodyweight compared between the presence of clinically relevant SRML.

		Total Protein Intake, g/kg				Total Energy Intake, kcal/kg		
Day	N	SRML	No-SRML	*p*-Value	N	SRML	No-SRML	*p*-Value
1	22/35	0 (0–0.18)	0.08 (0–0.19)	0.380	22/35	0 (0–3.54)	1.77 (0–3.43)	0.399
2	22/35	0.06 (0–0.71)	0.45 (0.10–0.81)	**0.029**	22/35	2.00 (0–13.58)	10.91 (2.50–14.37)	**0.042**
3	22/35	0.25 (0–1.04)	0.96 (0.28–1.20)	**0.029**	22/35	7.03 (0–19.08)	15.00 (5.20–21.21)	0.057
4	22/35	0.33 (0–1.25)	0.82 (0.19–1.59)	0.115	22/35	6.46 (0.33–20.94)	14.87 (3.99–26.14)	0.142
5	21/34	0.32 (0.02–1.16)	0.99 (0.41–1.55)	**0.019**	21/34	6.52 (1.50–21.29)	18.47 (5.78–28.25)	**0.030**
6	21/33	0.40 (0.23–1.54)	1.16 (0.53–1.85)	0.087	21/33	8.99 (4.35–28.19)	18.20 (10.01–32.00)	0.072
7	19/31	0.50 (0.20–1.29)	1.09 (0.60–1.65)	0.137	19/31	11.55 (3.95–21.07)	17.62 (9.34–29.13)	0.134

Data are presented as median (IQR). N = number of patients with available nutritional diaries per postoperative day; SRML = surgery-related muscle loss.

## Data Availability

The datasets generated and/or analyzed during the current study are not publicly available considering the data are linked to a vulnerable patient population but are available from the corresponding author on reasonable request.

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
