# Peer review of "Surgery-Related Muscle Loss after Pancreatic Resection and Its Association with Postoperative Nutritional Intake"

_cancers, 2023, doi:10.3390/cancers15030969_

Round 1

Reviewer 1 Report

Dear author,
I read with interest your study.
This article is part of an interesting project (MUSCLE POWER study) and its protocol has been previously published.
The preoperative data are well documented.
However:
- are you able to add albumin and pre-albumibne (table 1)?
- I think that morbidity and mortality should be given at 90 days. Even though the length of hospitalization is relatively short (12; 10-17).
Mortality at 30 days is 0. It would be interesting to confirm this information at 90 days, as for C/D, by detailing the reasons for reH (angiocholitis, radiological drainage etc. etc.) and reintervention (table 2).

This is a monocentric, descriptive, prospective cohort study, over a small period and with a relatively small sample. Partly explained because of COVID.
- Was there an evaluation of the NSN (number of subjects needed)?
- The histological and surgical heterogeneity may condition the results of the analyses.
Thank you for arguing this choice and including it in the limitations of the study.

Concerning the main objective.
- Why was the seventh postoperative day chosen?  
Are there any data in the literature?
Thank you for arguing this choice.

- Methodologically, wouldn't it be interesting to perform and add a uni and multivariate regression?

- In the results and conclusions (correlation/Pearson and variables of interest/UV and MV regression to be added), given the sample size and the heterogeneity of the population, I think that the results should be presented in a descriptive way.
These results probably need further study (MUSCLE POWER), but, to my knowledge, compared to the literature (sarcopenia/psoas diameter measurement) are surprising.

Sincerely

Reviewer 2 Report

Surgery related muscle loss after pancreatic resection and its association with postoperative nutritional intake- Hogenbrink et al.

COMMENTS TO THE AUTHORS:

Thank you for giving me the chance to review this manuscript which addresses a clinically relevant topic. Hogenbrink et al. performed a thorough analysis on clinically relevant surgery related muscle loss after pancreatic surgery.

Overall:

·         Very well written with clear analysis and overall presentation of the results.

Abstract:

·         No comments

Introduction:

·         No comments

Methods:

·         Line 124-125: decline of >=10% is clinically relevant? Reference?

·         Line 144: please define clinically relevant

Results:

·         Please consider creating a dot figure of the muscle mass assessments on day prior to surgery and day 7

·         Line 232: please comment in the methods or discussion what the indications were for enteral feeding tube after surgery and how it affected results

Discussion:

·         Please also comment on the potentials of preoperative nutritional support during e.g. neoadjuvant therapy or preoperative optimalisation as 44% of patients have preoperative weight loss >5%

·         Please explain the external validity of the results (neoadjuvant therapy?)

·         Lines 270-280: please comment on different muscles used for measurements, total abdominal, psoas and current study

Again, thank you for the invitation to review this paper.

Reviewer 3 Report

The study design is not appropriate to answer the research.A lot of informations about serolagical exams are lacking (serum albumin and proteinemia in pre and post operative time etc.). Type of operations considered for the study (pylorus pre-serving pancreatoduodenectomy [PPPD], Whipple’s procedure, distal pancreatectomy and total pancreatectomy)may not considered having the same  postoperative risk  of complications, outcome and patient functional recovery and surgery related muscle loss

Round 2

Reviewer 1 Report

Dear author,
thank you for making the necessary changes in the manuscript and enriching the tables.
In the present form the study can be, in my opinion, accepted in a first descriptive phase and we are looking forward to the results of MUSCLE POWER.

Sincerely

Reviewer 3 Report

Secondary endpoints about complication rate, length of hospital stay, readmission rate and  SRML should be discussed and an  effort  for better understanding about these results should be attempted.